# Metal coordination and enzymatic reaction of the glioma-target R132H isocitrate dehydrogenase 1: Insights by molecular simulations

**Bharath Raghavan**[1,2¤], **Marco De Vivo**[3], **Paolo Carloni**[1,4*]

1 Computational Biomedicine, Institute for Neuroscience and Medicine 9, Forschungszentrum Jülich GmbH, Jülich, Germany, 2 Department of Physics, Rheinisch-Westfälische Technische Hochschule Aachen University, Aachen, Germany, 3 Molecular Modelling and Drug Discovery, Italian Institute of Technology, Genova, Italy, 4 Department of Physics and Universitätsklinikum, Rheinisch-Westfälische Technische Hochschule Aachen University, Aachen, Germany

¤ Current address: National Center for Computational Sciences, Oak Ridge National Laboratory, Oak Ridge, United States of America

* p.carloni@fz-juelich.de

**Data availability statement:** Input files, coordinates and trajectories related to the MiMiC QM/MM MD simulations of mut-IDH1

## Abstract

R132H IDH1 is an important therapeutic target for a variety of brain cancers, yet drug leads and radiotracers which selectively bind only to the mutant over the wild type are so far lacking. Here we have predicted the structural determinants of the Michaelis complex of this mutant using a QM/MM MD-based protocol. It shows some important differences with the X-ray structure, from the metal coordination to the positioning of key residues at the active site. In particular, one lysine residue (K212) emerges as a mostly likely proton donor in the key proton-transfer step of the R132H IDH1 catalytic reaction. Intriguingly, the same residue in its deprotonated state is likely to be involved in the reaction catalyzed by the wild-type enzyme (though the mechanisms are different). Our QM/MM protocol could also be used for other metal-based enzymes, which cannot be modelled easily by force field-based MD, like in this case.

## Introduction

The NADP-dependent, magnesium-based human Isocitrate Dehydrogenase 1 (IDH1) homodimeric enzyme catalyzes the oxidative decarboxylation of isocitrate (ICT) to α-ketoglutarate (α KG, Eq 1) [1–3].

$$ICT + NADP^+ \rightarrow \alpha KG + NADPH \tag{1}$$

The α KG product regulates the behavior of many dioxygenases enzymes in humans, like the hypoxia-inducible factor-1α and the ten-eleven translocation DNA hydroxylases [4,5]. This has a direct impact on cell stemness and differentiation. Unfortunately, several mutations of this enzyme are involved in a variety of brain cancers. Particularly important

are publicly available on Zenodo (DOI: 10.5281/zenodo.13862463). The code for MiMiC is made publicly available at: https://gitlab.com/mimic-project.

**Funding:** This work was funded by the Helmholtz European Partnering program ("Innovative high-performance computing approaches for molecular neuromedicine") The funders had no role in study design, data collection and analysis, decision to publish, or preparation of the manuscript.

**Competing interests:** The authors have declared that no competing interests exist.

is the Arg132His variant (mut-IDH1 hereafter), associated with astrocytoma and oligodendroglioma progression [6–9].

Mut-IDH1, in addition to the reaction in Eq 1 [10], catalyses the conversion of α KG to 2-hydroxyglutarate (2-HG, Eq 2) [11].

$$\alpha KG + NADPH \rightarrow 2 - HG + NADP^+ \tag{2}$$

2-HG is a known oncometabolite that promotes stemness in human cells and inhibits DNA demethylases [12,13]. X-ray studies on the $Ca^{2+}$-substituted enzyme show that both wt-IDH1 [3] and mut-IDH1 [11] are dimers (Fig 1A), with each of the two monomers mostly catalytically independent [14]. The NADP(H) cofactor, the substrate and the metal ion are located in each active sites. The latter include residues from both monomers (In this text, residues from the second subunit are labelled by a dash and those from the first subunit are left unmarked.).

A model of wt-IDH1 Michaelis complex has been obtained from molecular simulation by us and Maria Ramos' group [15,16], based on the available X-ray structure [3]. Arg100, Arg109, Lys212', Tyr139, Thr75 and Ser94 bind ICT to the active site (Fig 1B). Here, Lys212' (in its deprotonated form) is the most likely residue that initiates the catalysis as a base (Fig 2A).

For mut-IDH1, indirect information on the Michaelis complex can be obtained by inspection of the X-ray structures (Fig 2B) [11]. The $Ca^{2+}$ ion is heptacoordinated. The substrate forms H-bonds with several of the same residues as ICT (Arg100, Arg109, Thr77, Ser94, and Asn96). The mutated residue His132 does not interact with the substrate. Most importantly, the nature of the interaction between α KG and the remaining residues (Lys212', Tyr139 or Asp275) depend on the protonation states of Lys212' and Asp275. In this text, we refer to the mut-IDH1 protomer with Lys212' deprotonated/Asp275 protonated as **K/D$^H$**, and the other way around with **K$^H$/D**. We should mention that all X-ray structures of mut-IDH1 (as well as the wild type) complexed with the substrate and cofactor, are catalytically inactive as the $Ca^{2+}$

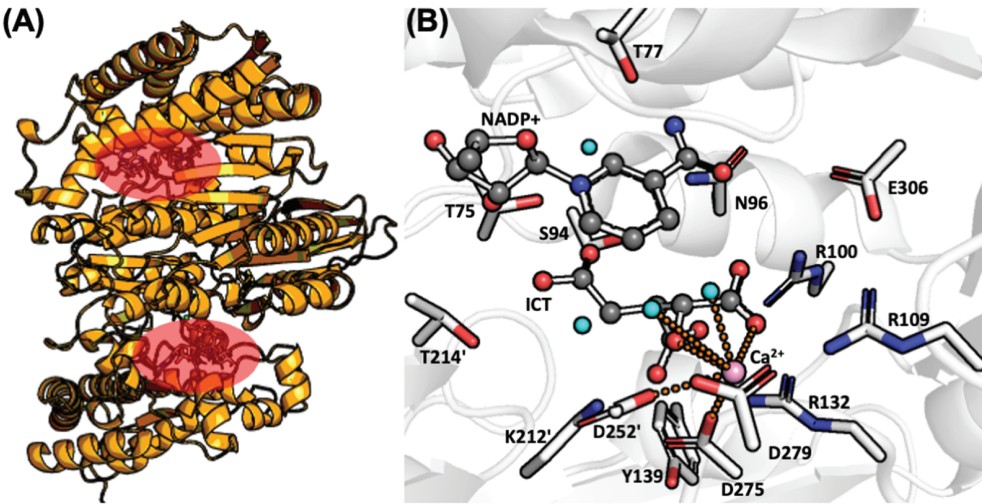

**Fig 1. (A) The dimeric structure of IDH1. The two active sites are shown in red.** (B) Representation of the wt-IDH1 active site from the X-ray structure. ICT and NADP$^+$ are shown in ball-and-sticks representation, while the protein residues are shown as sticks. The $Ca^{2+}$ ion is shown in pink.

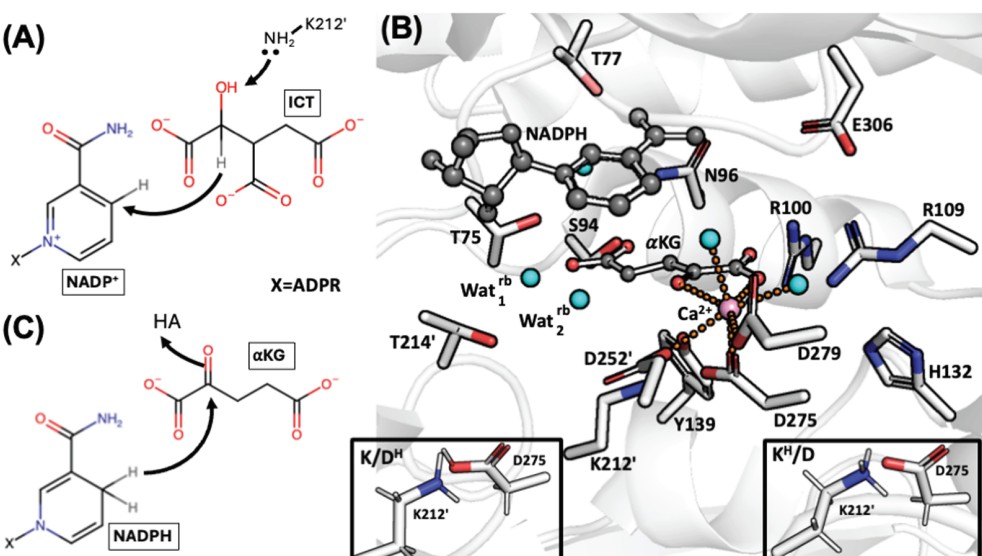

**Fig 2. (A) The first step of the oxidative decarboxylation of ICT to α KG as catalysed by wt-IDH1, with Lys212' as the bse initiator** [16]. (B) Representation of the mut-IDH1 active site from the X-ray structure. α KG and NADPH are shown in ball-and-sticks representation, while the protein residues are shown as sticks. The $Ca^{2+}$ ion (in pink) binds to the α-carboxylate of α KG, the α KG of ICT, Asp275, Asp252', Asp279 and two water molecules. The nicotinamide group of the NADPH cofactor forms H-bonds Glu306 and Asn96. The ribose alcohol of the cofactor is anchored to α KG by two water molecules, $Wat^{rb}_1$ and $Wat^{rb}_2$. The former forms H-bonds with the γ-carboxylate of α KG, $Wat^{rb}_2$, Thr214' and the ribose. $Wat^{rb}_2$ in turn, interacts with the αketone of α KG. Tyr139 H-bonds to Asp275, which in turn forms a salt bridge with Lys212'. Insets of the **K/D$^H$** and **K$^H$/D** protomers are also shown. (C) Proposed reaction mechanism for α KG to 2-HG as catalyzed by mut-IDH1. Here the acid from the protein, HA, is unknown.

ion replaces the $Mg^{2+}$ ion [3,11,17–19]. This obviously may lead to a different coordination and hence a different structure of the active site.

The conversion of α KG to 2-HG by mut-IDH1 can also be compared to that of pyruvate to lactate by lactacte dehydrogenase (LDH) [20,21]. Both involve reduction of a ketone to alcohol by an acidic residue from the protein, and donation of hydride by nicotinamide ring (NADPH in mut-IDH1 and NADH in LDH). The catalytic mechanism of the enzyme has been very recently studied by quantum chemistry methods [10]. However, an important issue has not been anddressed so far: the protein residue performing the proton transfer to the α-ketone of α KG (Fig 2C) has not been established. This severely limits our understanding of the mechanism.

Inhibitors which selectively bind mut-IDH1 and not wt-IDH1, so far lacking [22,23], could be excellent drug leads against glioma. They could also work as positron emission tomography (PET) biomarkers (or radiotracers) [24] for glioma progression by non-invasively and selectively detecting mut-IDH1 expression (PET is a molecular imaging technique which uses specific probes that are labeled with positron-emitting radioisotopes to visualize and measure changes in biological processes in vivo. These probes are referred to as radiotracers, and are labeled with [18]F radioactive fluorine isotopes [18]F.). Unfortunately, all current proposed ligands do not bind competitively at the substrate site, but rather at the dimer interface [23,25]. This region is structurally similar in both mut- and wt-IDH1, and thus the ligands are not binding-selective [23].

The drug design efforts for binding-selective inhibitors might greatly profit from the structure of the Michaelis complexes of wt-IDH1 (available from previous QM/MM studies [15, 16]) and mut-IDH1 (still lacking). Performing structure-based drug design solely using the inactive crystal structure might lead to incorrect results due to the differences between Ca(II) and Mg(II) coordination chemistry.

Here, based on the X-ray structure of the inactive mut-IDH1/Ca$^{2+}$ complex (Fig 2B), we have predicted the structural determinants of the active mut-IDH1/Mg$^{2+}$ complex by using the massively parallel and flexible MiMiC-QM/MM approach developed recently by a large consortium including some of us [26,27]. Classical MD simulations of this protein did not provide a correct geometry of the Mg$^{2+}$ coordination in the active site. To obtain the correct coordination sphere, a new QM/MM MD protocol was developed (Fig 3B). This involved an initial minimization of the structure at the QM/MM level, and 80 ps of QM/MM MD of mut-IDH1. This provides the first insight on active mut-IDH1 for drug design and allows us to suggest that protonated Lys212' is the most likely proton donor in the catalysis.

## Methods

### General parameters and setup

**Initial models.** We used the human mut-IDH1 enzyme homodimeric X-ray structure, containing two identical active sites featuring a Ca$^{2+}$ ion, the α KG substrate and the NADPH cofactor (PDB code: 3INM, Fig 2B) [11]. First, the Ca$^{2+}$ ion was replaced with a Mg$^{2+}$ ion. Second, we added the N- and C- terminus (Met1 to Lys4 and Ala410 to Leu414), lacking in the X-ray structure, using the Modeller code [28]. Hydrogens were added to the protein using the *pdb2gmx* command in GROMACS [29]. Both the **K/D**$^{\mathbf{H}}$ and the **K**$^{\mathbf{H}}$**/D** protomers were considered (Fig 2B). They were placed in a cubic box of edge length 10.9 nm, and filled with ~3,800 water molecules, along with 14 sodium ions. The overall systems were neutral.

**Force-field based MD setup.** The Amber99sb*-ildn force field [30], TIP3P [31], and parameters from Ref [32] were used for the protein, water, and NADPH respectively. The bonded and van der Waals force field parameters for α KG were generated using the Generalised Amber Force Field [33]. The partial charges were calculated using the RESP method at the HF/6-31G* level of theory. This was done using the ANTECHAMBER software package and ACPYPE [34,35]. NVT MD simulations were achieved using the Nosé-Hoover thermostat [36] at 300 K. NPT MD simulations were achieved using the Parrinello-Rahman barostat [37] at 1 bar with a time constant of 2 ps. Restraints were added by using PLUMED version 2.8.1. [38,39]. The GROMACS version 2019.4 [40] code was used for all calculations.

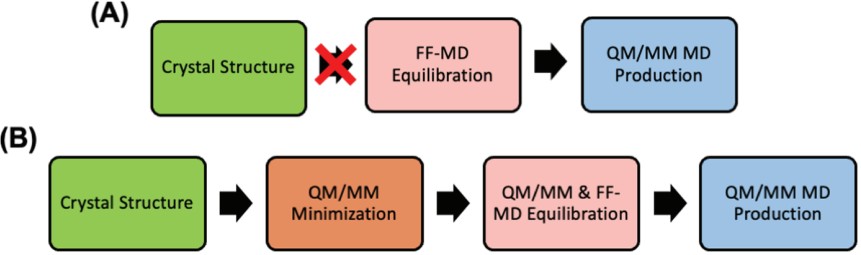

**Fig 3. Protocol (A) failed in predicting the Mg$^{2+}$ ion coordination.** Hence we used protocol (B).

**QM/MM MD setup.** The systems were divided into MM and QM subregions, using the MiMiCPy code [41]. In various step of the MD protocol (discussed later), the following QM subregions were considered (Fig 4A):

> **Group I** consisted of α KG, $Mg^{2+}$, the nicotinamide ring of NADPH, the water between NADPH ribose and α KG γ-carboxylate, and the enzyme residues around both the α-carboxylate of α KG and $Mg^{2+}$ were included: Arg100/109, His132, Tyr139, Lys212', Asp252/275/279, and the two water molecules coordinating with $Mg^{2+}$ (140 atoms).
>
> **Group II** in addition to the previous selection, residues and waters involved in binding to the γ-carboxylate of α KG were included. This consisted of Th77/75/214', Ser94, Asn96. The QM region differed slightly between the $K/D^H$ and $K^H/D$ protomers, as some of the residues interacting with α KG differed (179 to 188 atoms, details in the SI).

The MM part was described by the same force fields as in the force field-based MD simulations. In the QM region, the quantum problem was solved with density functional theory at the BLYP level [42]. The wavefunction was expanded using a plane-wave basis set up to a cutoff of 100 Ry. The core electrons were described using norm-conserving pseudopotentials [43]. The valence electrons were treated explicitly. The QM region was inserted in a cubic box of edge 2.43 nm. Isolated system conditions were achieved using the Tuckerman method of the Poisson solver [44].

Open valances at the boundary of the QM-MM covalent bonds were treated with monovalent pseudopotentials [45]. Electrostatic interactions between the QM and MM subsystems were described using the Hamiltonian electrostatic coupling scheme of Laio et al.[46]. The short-range QM-MM electrostatic interactions were computed explicitly within a cutoff radius of 1.69 nm while the long-range interactions between the point charges of the MM region and the QM charge density were computed using a 5th order multipole expansion of the QM electrostatic potential. QM/MM MD was run using the Born-Oppenheimer

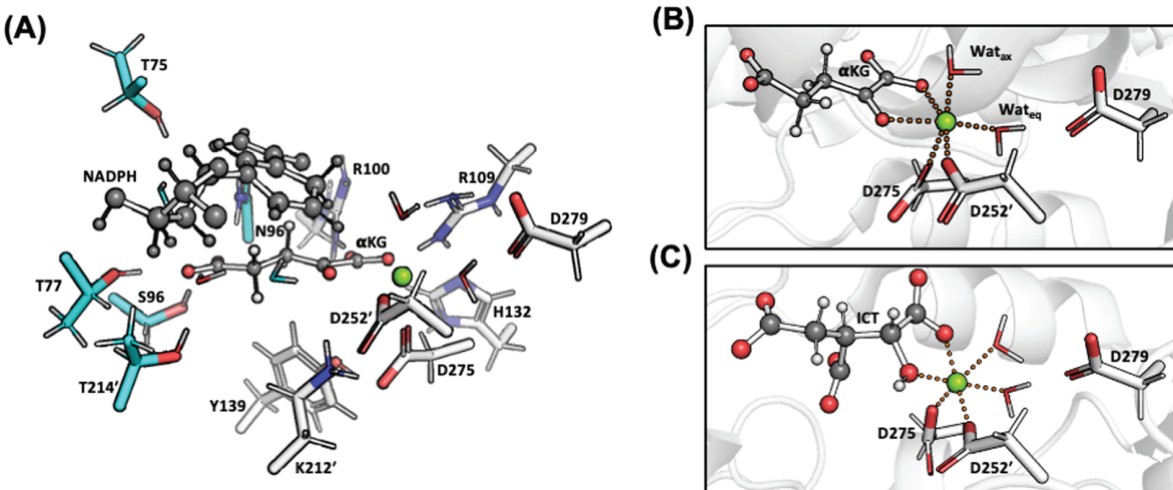

**Fig 4. (A) Group I (gray) and Group II (gray and blue) QM regions in the QM/MM calculations.** (B) Hexacoordination sphere of $Mg^{2+}$ in the $K^H/D$ protomer emerging from our QM/MM calculations. The same coordination is also obtained for the $K/D^H$ protomer. (C) Hexacoordination sphere of $Mg^{2+}$ in wt-IDH1 Michaelis complex obtained from molecular simulation by us [15].

approach, with a timestep of 0.5 fs. Temperature was maintained around 300 K using a Nosé-Hoover thermostat. Restraints were added by using PLUMED version 2.8.1, and constraints using the built-in algorithm in CPMD.

The setup and parameters used here are identical to those used by us for simulating the wt-IDH1 [15]. All calculations were carried out using GROMACS 2020.3 [40] and CPMD 4.3 [47] interfaced with MiMiC 0.2.0 [48] (including the MiMiC Communication Library 2.0.1 [49] for server-client communication).

## QM/MM MD protocol

MD simulations based on three different force field parameters for $Mg^{2+}$ ions [30,50,51] failed to reproduce the bidentate chelation of α KG with $Mg^{2+}$ (S1 Fig). This chelation should be maintained as it plays a crucial role in promoting the catalysis [11]. This points to difficulties that some force fields experience in describing the coordination chemistry of the metal ion in this complex enzyme (Fig 2A). Of course, it is entirely possible that other force fields, not used here, could provide more accurate results. Here, in order to overcome this problem, we decided to transcend the use of the force fields and use QM/MM simulations. These have shown to accurately reproduce the structural determinants of metalloproteins [52–55] (including wt-IDH1 [15]). We utilized the following protocol (Fig 3B):

(1) Energy Minimization.
  (A) Steepest descent minimization of the MM hydrogens.
  (B) 100 ps of heating from 0 K to 300 K of force-field MD of the water molecules and the sodium ions. The rest of the system was restrained with position restraints on heavy atoms.
  (C) 4,000 steps of QM/MM energy-minimization of the entire system. This is done by using simulated annealing, where the temperature is gradually decreased from 300 K to 0 K at a rate of 1% pre timestep. The QM region consisted of **Group I** (Fig 4A), while the remaining atoms were treated at the MM level. The active site of two subunits were alternatively treated at the QM level, while the $Mg^{2+}$ coordination sphere (α KG, Asp252/275/279, and the two water molecules) of the other subunit was kept constrained.
(2) Equilibration.
  (A) 2000 steps QM/MM MD heating from 0 to 300 K, followed by ∼8 ps of NVT QM/MM MD. Also here, The QM region consisted of **Group I** (Fig 4A). As in step 1(C), the active sites were alternatively treated at the QM level with the other constrained.
  (B) 5 ns NVT and 5 ns NPT force-field based MD, with position restraints on the non-hydrogen atoms.
  (C) 500 ns NPT MD without position restraints. Two key interactions at the active site were restrained: (i) the α-ketone of α KG / $Mg^{2+}$ coordination bond (upper wall restraint set at 2.3 Å), (ii) $Wat^{rb}_1$ / NADPH ribose, γ-carboxylate of α KG, and Thr214' (distance restraints set to ∼1.9 Å).
(3) Production. QM/MM MD of the entire system was performed starting from the last snapshot of step (2)C, using **Group II** as QM region. The two active sites were separately simulated at the QM level by 20 ps QM/MM MD. The other subunit was treated at the MM level with the same restraints as in step (2)C. This was repeated for both Lys212'-H/Asp275 and the Lys212'/Asp275-H protomers, resulting in a total of 80 ps with of QM/MM MD. The last 10 ps for each simulation were used for analysis.

This procedure allowed for the prediction of the $Mg^{2+}$ coordination sphere. In contrast to the force-field based MD, here the metal coordination turned out to be similar to that of the wt-IDH1 (see Results and Discussions).

## Results

We used our QM/MM MD protocol (Fig 3B) to predict the structural determinants of both **$K/D^H$** and **$K^H/D$** protomers. Both active sites A and B were simulated alternatively at the DFT-BLYP QM level, for a total of 80 ps QM/MM MD. Because of the high-scalability of the MiMiC code used for QM/MM simulations [15,26,56,57], this took only ~2 weeks on the JUWELS machine in the Jülich Supercomputing System [58]. The deviation of the heavy atoms in the QM region from the X-ray structure at the end of the QM/MM MD is between 1 and 2 Å (S3 File). Below, we describe our predicted models in detail, starting from a feature which turns out to be the same across all the systems studied: the type of $Mg^{2+}$ ion coordination.

**$Mg^{2+}$ Coordination**. In all circumstances, Asp279 does not bind directly to the metal ion as it does to $Ca^{2+}$ in the X-structure. As a result, the Mg(II) ion is hexacoordinated across all of the systems studied here (Fig 4B): namely, this metal ion binds to the α-ketone and α-carboxylate groups of α KG, to the side chains of Asp275 and Asp252', and to equatorial ($Wat_{eq}$) and axial waters ($Wat_{ax}$). $Wat_{ax}$ further H-bonds with Asp279. The metal coordination sphere is essentially the same as that in the wt-IDH1/ICT complex, except of course, that the α-alcohol of ICT is replaced by the α-ketone of α KG (Fig 4C). The distances of the coordinating atoms from $Mg^{2+}$ in the metal coordination bonds range between 2.0–2.2 Å (Table 1), except for that between $Mg^{2+}$ and the α-ketonic oxygen. This latter varies with the protomer used, as discussed below.

**$K^H/D$**. In both active sites, the α-ketonic oxygen of α KG interact not only with the $Mg^{2+}$ ion, but also forms weak H-bonds to Lys212' and $Wat^{rb}_2$ (Table 2 and Fig 5A). As in the X-ray structure, $Wat^{rb}_2$ forms a H-bond with $Wat^{rb}_1$, which in turn, interacts with Thr214' and NADPH. Lys212' also forms salt bridges with Asp252' and Asp275. The latter further forms a H-bond with Tyr139. In active site A and the X-ray structure, Arg100/109 bind the the α-carboxylate of α KG (as in the X-ray structure), while in active site B this is broken (Fig 5A). Asn96 forms an H-bond with the α KG γ-carboxylate in active site A (as in the X-ray structure), however this Ans96 moves to bind with the α-carboxylate in active site B.

**Table 1. Average distance (in Å) from the QM/MM MD simulation of mut-IDH1 between $Mg^{2+}$ and the atoms in its coordination sphere. The α-ketonic oxygen of α KG is not included, as it is given in Table 2.**

| Protomer | Active Site | α KG α-carboxy | Asp275 | Asp252' | $Wat_{ax}$ | $Wat_{eq}$ |
|---|---|---|---|---|---|---|
| $K^H/D$ | A | 2.1±0.1 | 2.0±0.1 | 2.0±0.1 | 2.2±0.1 | 2.1±0.1 |
| | B | 2.1±0.1 | 2.1±0.1 | 2.1±0.1 | 2.1±0.1 | 2.1±0.1 |
| $K/D^H$ | A | 2.2±0.1 | 2.1±0.1 | 2.1±0.1 | 2.1±0.1 | 2.1±0.0 |
| | B | 2.2±0.1 | 2.1±0.1 | 2.1±0.1 | 2.2±0.1 | 2.1±0.1 |

**Table 2. Average distance (in Å) from the QM/MM MD simulation of mut-IDH1 between the α-ketonic oxygen of α KG and its interaction partners.**

| Protomer | Active Site | $Mg^{2+}$ | Lys212' | $Wat_{Y139}$ | $Wat^{rb}_2$ |
|---|---|---|---|---|---|
| $K^H/D$ | A | 2.4±0.2 | 2.2±0.2 | — | 2.5±0.3 |
| | B | 2.6±0.2 | 2.1±0.1 | — | 2.7±0.6 |
| $K/D^H$ | A | 2.3±0.1 | — | 2.4±0.2 | 2.7±0.2 |
| | B | 2.2±0.1 | — | — | 2.3±0.2 |

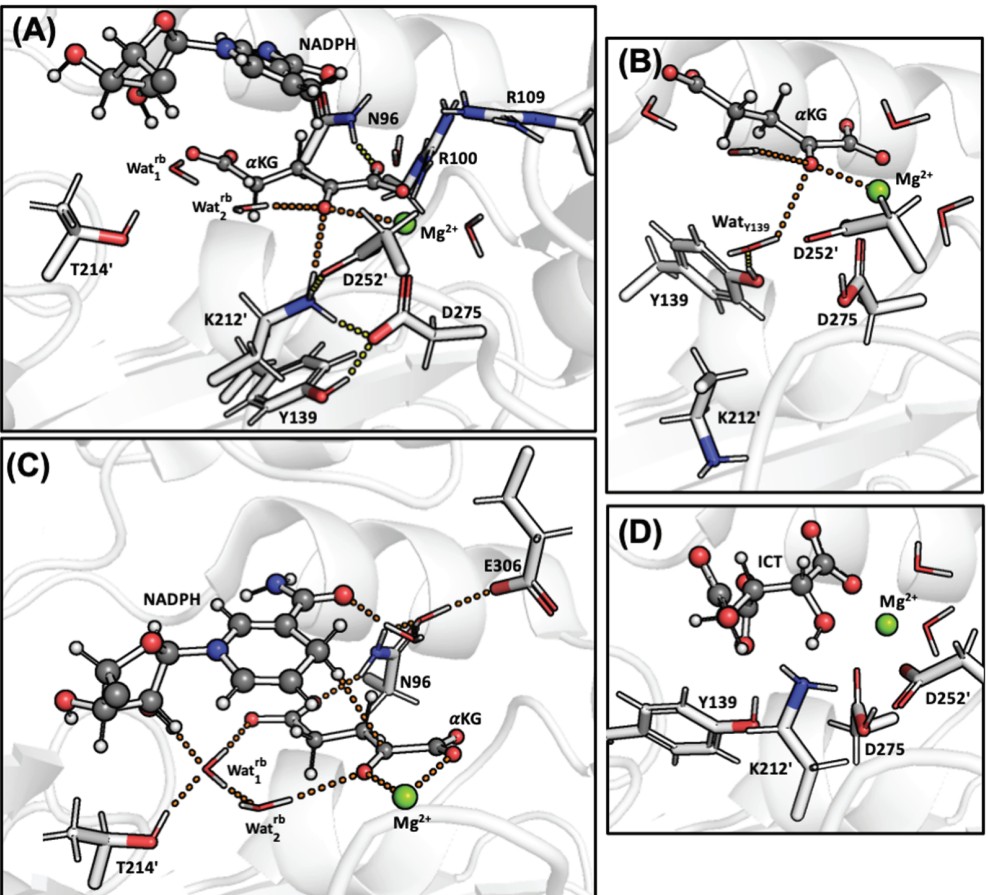

**Fig 5. Snapshots of mut-IDH1 after 20 ps of QM/MM MD: (A) Active site B of the $K^H/D$ protomer showing interactions of the α-ketone of α KG (in orange), and those of Lys212', Tyr139, and Asn96.** (B) Active site A of the $K/D^H$ protomer showing the interactions of the α-ketone of α KG (in orange), especially with $Wat_{Y139}$. (C) Active site B of the $K/D^H$ protomer showing the interactions that anchor NADPH close to α KG (in orange). (D) Snapshot of ICT–Lys212' (deprotonated) in the wt-IDH1 active site from Ref [16].

$K/D^H$. In both active sites, the α-ketone of α KG binds to the $Mg^{2+}$ ion (Table 2); Lys212' is rotated away from the α-ketone (Fig 5B). It also interacts weakly with $Wat^{rb}_2$, though this interaction is stronger in active site B. As in the X-ray structure, $Wat^{rb}_2$ forms a H-bond with $Wat^{rb}_1$, which in turn, interacts with Thr214' and NADPH. Tyr139 establishes a water mediated H-bond with His132 in active site B, and, at times, a water-mediated interaction with the α-ketone of α KG in active site A (Fig 5B and Table 2, where this water is refered to as $Wat_{Y139}$; also see SI). Arg100 and Arg109 form H-bonds with α-carboxylate of α KG. Ans96 H-bonds with the γ-carboxylate and forms a water-mediated interaction with NADPH (Fig 5C). The water in the latter interaction is also stabilized by Glu306. These interactions, also present in $K^H/D$, anchor the cofactor to the substrate. Only in active site B of $K^H/D$, is the interaction with Asn96 and γ-carboxylate broken (compare Fig 5A with 5C).

## Discussion

The goal of this work was to obtain the structural determinants of mut-IDH1 Michaelis complex. Attempts to use a variety of force fields (as per the protocol in Fig 3A) did not lead to

satisfactory results. We thus resorted to a new QM/MM MD protocol (Fig 3B), which turned out to be more successful.

Our calculations suggest that, irrespective of the protonation state of Lys212' and Asp275, the metal ion is hexacoordinated. The ion binds to Asp275/252', two water molecules, and to the substrate (Fig 4B). This contrasts with the $Ca^{2+}$ in the X-ray structure, which is heptacoordinated (Fig 2B). This is likely one important reason why the enzyme is inactivated by $Ca^{2+}$ ions. The magnesium coordination sphere in the α KG/mut-IDH1 complex is similar to that of the ICT/wt-IDH1 complex (Fig 4C), except that the substrate-$Mg^{2+}$ binding in wt-IDH1 is much stronger. The arrangement around the magnesium obtained here is very similar to that obtained from previous quantum chemical studies of the reaction [10]. The α KG–$Mg^{2+}$ binding distance, especially in $K^H/D$, is more elongated than the expected distance of 2.1 Å (Table 2). This is likely due to the extra proton on Lys212' pulling the α-ketone of α KG away from the $Mg^{2+}$ coordination sphere. Our result is thus consistent with the claim by a previous study that states that the α KG–$Mg^{2+}$ binding in mut-IDH1 is weaker than the ICT-$Mg^{2+}$ binding in wt-IDH1 [59]. Such a weak interaction between α KG and $Mg^{2+}$ may be difficult to describe by standard force fields (as suggested by our own work here), which, in the way they are built, may assume an ideal octahedral geometry.

A second difference with the X-ray structure [11] is the location of residues which can act as proton donor to the α-ketone of α KG in the enzymatic reaction (Fig 5A). Inspection of the catalytically inactive X-ray structure show that Lys212', Asp275 and Tyr139 interact with the ketone moiety of the substrate. However according to our calculations, Lys212' in its positively charged (protonated) state is the most likely residue to protonate the substrate, as it is the only one forming an H-bond with the α-ketone in the QM/MM Michaelis complex. The other two are at least as far as 4 Å from the α-ketone of α KG (Fig 5B and S2C Fig). This finding is consistent with mutagensis experiments, which show that Y139D-IDH1 is still able to catalyze the reaction [17]. The usage of QM/MM dynamics in this work (as opposed to the single-point energy calculations) allows us to propose this result. Interestingly, Lys212' (in its deprotonated state) is involved in the deprotonation of the substrate (ICT) in the wt-IDH1 [15,16]. Thus, the same residue appears to be well positioned to perform proton transfers in both wt- and mut-IDH1 isoforms (compare the position of Lys212' vis á vis the substrate in Fig 5A and 5D). This information is not provided by previous quantum chemical studies of the reaction [10]. However, while the simulations do suggest a key role for Lys212' our calculations do not rule out that other residues (such as, for instance, Tyr139), under different solvent and/or pH conditions, could act as proton donors in the enzymatic reaction.

## Conclusions

Developing binders selective to mut-IDH1 over wt-IDH1 could have wide-ranging applications, from PET radiotracers for early, non-invasive diagnosis of IDH1-associated glioma to anti-brain cancer drugs. Together with our previous work, we have provided the structural determinants of wt- and mut-IDH1. The latter structure show crucial differences with the X-ray structure, from the metal coordination to the positioning of key residues at the active site. In particular, our predictions further allow us to suggest that Lys212' is the mostly likely proton donor in the mut-IDH1 catalytic reaction. The same residue (deprotonated) is also important in the wt-IDH1 catalysis as a proton acceptor. Thus, it appears that Lys212' performs the required proton transfers in both wt- and mut-IDH1 isoforms. Importantly, our structures may be used as templates for the design of binding-selective ligands, which unfortunately have not been identified yet.

The input files, and QM/MM trajectories are made publicly available with this work and can be used in a drug design protocol for suggesting radiotracer precursor candidates of mut-IDH1.

## Supporting information

**S1 Fig. (A) Cartoon representation of the mut-IDH1 with the active site containing the heptacoordinated $Ca^{2+}$ coordination sphere.** The α KG substrate is coordinated to $Ca^{2+}$ in a bidendate fashion. (B) Loss of the bidendate coordination of α KG during MD simulations based on the Amber99sb*-ildn force field [60,61].
(PDF)

**S2 Fig. The probability distributions of the angle vs distance for Lys212'–α-ketone of α KG interactions (A) in active site A, and (B) in active site B of the $K^H$/D protomer, (C) the α-ketone of α KG and Tyr139-water in active site A of the K/$D^H$ protomer.** Because of its geometry, this last interaction cannot be considered as an H-bond. The distributions have been calculated as a kernel-density estimate using Gaussian kernels, starting from 10 ps of the QM/MM MD.
(PDF)

**S3 File. Supplementary text.** Includes a description of the classical force-fields explored, various QM/MM regions used and further discussion on the QM/MM dynamics.
(PDF)

## Acknowledgments

BR and PC acknowledge the Gauss Centre for Supercomputing e.V. (www.gauss-centre.eu) for providing computing time through the John von Neumann Institute for Computing (NIC) on the GCS Supercomputer JUWELS [58] at Jülich Supercomputing Centre (JSC). BR also gratefully acknowledges discussions with Emiliano Ippoliti, Davide Mandelli and Florian K. Schackert from Forschungszentrum Jülich.

## Author contributions

**Data curation:** Bharath Raghavan.

**Formal analysis:** Bharath Raghavan.

**Funding acquisition:** Marco De Vivo, Paolo Carloni.

**Investigation:** Bharath Raghavan.

**Methodology:** Bharath Raghavan.

**Project administration:** Marco De Vivo.

**Supervision:** Marco De Vivo, Paolo Carloni.

**Validation:** Bharath Raghavan.

**Visualization:** Bharath Raghavan.

**Writing – original draft:** Bharath Raghavan.

**Writing – review & editing:** Paolo Carloni.

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
