## [Decision Letter · Decision Letter 0]

29 Mar 2025

PONE-D-24-47203Metal Coordination and  Enzymatic Reaction  of the Glioma-Target R132H Isocitrate Dehydrogenase 1: Insights  by  Molecular SimulationsPLOS ONE

Dear Dr. Carloni,

Thank you for submitting your manuscript to PLOS ONE and sorry for the delays. After careful consideration, we feel that it has merit but does not fully meet PLOS ONE’s publication criteria as it currently stands. Therefore, we invite you to submit a revised version of the manuscript that addresses the points raised during the review process.

No new experiments are required, but please address reviewer #2's class to consider other "donors".

We look forward to receiving your revised manuscript.

Kind regards,

Michael Klymkowsky, Ph.D.

Academic Editor

PLOS ONE

Journal Requirements:

This work was funded by the Helmholtz European Partnering program ("Innovative high-performance computing approaches for molecular neuromedicine")

6. We noted in your submission details that a portion of your manuscript may have been presented or published elsewhere. The preprint of the manuscript was submitted:https://doi.org/10.26434/chemrxiv-2024-jxd73

Parts of the results were also published as Chapter 7 of the thesis by Bharath Raghavan (first author): https://doi.org/10.18154/rwth-2024-07554. Please clarify whether this [conference proceeding or publication] was peer-reviewed and formally published. If this work was previously peer-reviewed and published, in the cover letter please provide the reason that this work does not constitute dual publication and should be included in the current manuscript.

Reviewers' comments:

Reviewer's Responses to Questions

**Comments to the Author**

1. Is the manuscript technically sound, and do the data support the conclusions?

Reviewer #1: Yes

Reviewer #2: Yes

2. Has the statistical analysis been performed appropriately and rigorously? 

Reviewer #1: N/A

Reviewer #2: N/A

3. Have the authors made all data underlying the findings in their manuscript fully available?

Reviewer #1: Yes

Reviewer #2: Yes

4. Is the manuscript presented in an intelligible fashion and written in standard English?

Reviewer #1: Yes

Reviewer #2: Yes

5. Review Comments to the Author

Reviewer #1: Here the Carloni team addresses a highly relevant and acute problem: the R132H mutation in Isocitrate Dehydrogenase 1 (IDH1) is a vital target in multiple brain cancers. IDH1 links metabolism to epigenetics, and as such plays a crucial role in these cancers, causing widespread changes in the epigenetic landscape of these tumors, through histone and DNA demethylases. IDH1/2 mutations can lead to hypermethylated histones and DNA which changes gene expression, commonly by overexpression. Despite the recognition of their critical tumorigenic role, still no drug leads and radiotracers which selectively bind only to the mutant over the wild type are so far lacking.

Here, through comprehensive computational analysis, the authors predicted the structural determinants of the Michaelis complex of this mutant using QM/MM MD. They observe important differences from the X-ray structure, ranging from metal coordination to the positioning of key residues at the active site. Especially, the K212 mutation appears as the likely proton donor in the key proton-transfer step of the R132H IDH1 catalytic reaction. The authors then discover that the same residue in its deprotonated state appears involved in the reaction catalyzed by the wild-type enzyme, albeit via different mechanisms.

The rigor of the work, its implications to drug discovery, and the new protocol that they employ, make me suggest accepting the manuscript as is. This is reenforced by the clarity of the presentation, text and figures.

Reviewer #2: This manuscript presents a QM/MM-based molecular simulation study of the active-site architecture and mechanistic features of the R132H mutant of human isocitrate dehydrogenase 1 (IDH1). By comparing their simulated structure to both the wild-type and the inactive crystal structure of the mutant, the authors make a case that the proton donor in the mutant’s reaction is most likely Lys212'—a result with potential implications for selective drug design. Overall this is a well-designed and executed study with a number of strengths including the biological and therapeutic relevance and along with the open availability of all the QM/MM input files and trajectories is commendable. The one minor concern of this reviewer is that the was little exploration of alternative proton donors: While the simulations suggest Lys212’ is a key proton donor, it might be helpful for readers if the authors at least mentioned the other potential donors (e.g., Tyr139, Asp275) under different solvent or pH conditions could be critical. While presenting additional free energy or reaction-path calculations could further solidify the argument that Lys 212 is critical, this is not necessary for publication.

6. PLOS authors have the option to publish the peer review history of their article (what does this mean?). If published, this will include your full peer review and any attached files.

Reviewer #1: No

Reviewer #2: No

---

## [Author Response · Author response to Decision Letter 1]

26 Apr 2025

Reply to the reviewers.

Reviewer #1: Here the Carloni team addresses a highly relevant and acute problem: the R132H mutation in Isocitrate Dehydrogenase 1 (IDH1) is a vital target in multiple brain cancers. IDH1 links metabolism to epigenetics, and as such plays a crucial role in these cancers, causing widespread changes in the epigenetic landscape of these tumors, through histone and DNA demethylases. IDH1/2 mutations can lead to hypermethylated histones and DNA which changes gene expression, commonly by overexpression. Despite the recognition of their critical tumorigenic role, still no drug leads and radiotracers which selectively bind only to the mutant over the wild type are so far lacking.

Here, through comprehensive computational analysis, the authors predicted the structural determinants of the Michaelis complex of this mutant using QM/MM MD. They observe important differences from the X-ray structure, ranging from metal coordination to the positioning of key residues at the active site. Especially, the K212 mutation appears as the likely proton donor in the key proton-transfer step of the R132H IDH1 catalytic reaction. The authors then discover that the same residue in its deprotonated state appears involved in the reaction catalyzed by the wild-type enzyme, albeit via different mechanisms.

The rigor of the work, its implications to drug discovery, and the new protocol that they employ, make me suggest accepting the manuscript as is. This is reenforced by the clarity of the presentation, text and figures.

We thank the reviewer for highly praising our manuscript.

Reviewer #2: This manuscript presents a QM/MM-based molecular simulation study of the active-site architecture and mechanistic features of the R132H mutant of human isocitrate dehydrogenase 1 (IDH1). By comparing their simulated structure to both the wild-type and the inactive crystal structure of the mutant, the authors make a case that the proton donor in the mutant’s reaction is most likely Lys212'—a result with potential implications for selective drug design. Overall this is a well-designed and executed study with a number of strengths including the biological and therapeutic relevance and along with the open availability of all the QM/MM input files and trajectories is commendable.

We are grateful for the reviewers' careful consideration of our work and their constructive feedback.

The one minor concern of this reviewer is that the was little exploration of alternative proton donors: While the simulations suggest Lys212’ is a key proton donor, it might be helpful for readers if the authors at least mentioned the other potential donors (e.g., Tyr139, Asp275) under different solvent or pH conditions could be critical. While presenting additional free energy or reaction-path calculations could further solidify the argument that Lys 212 is critical, this is not necessary for publication.

We now add the following sentence (in quotes) to the last sentence in the discussion:

Thus, the same residue appears to be well positioned to perform proton transfers in both wt- and mut-IDH1 isoforms (compare the position of Lys212' vis a vis the substrate in Figs 5A and D). This information is not provided by previous quantum chemical studies of the reaction. [10] "However, while the simulations do suggest a key role for Lys212’, our calculations do not rule out that other residues (such as, for instance, Tyr139), under different solvent and/or pH conditions, could act as proton donors in the enzymatic reaction"

---

## [Decision Letter · Decision Letter 1]

30 May 2025

Metal Coordination and  Enzymatic Reaction  of the Glioma-Target R132H Isocitrate Dehydrogenase 1: Insights  by  Molecular Simulations

PONE-D-24-47203R1

Dear Dr. Carloni,

We’re pleased to inform you that your manuscript has been judged scientifically suitable for publication and will be formally accepted for publication once it meets all outstanding technical requirements.

Kind regards,

Michael Klymkowsky, Ph.D.

Academic Editor

PLOS ONE

Additional Editor Comments (optional):

Reviewers' comments:

Reviewer's Responses to Questions

**Comments to the Author**

1. If the authors have adequately addressed your comments raised in a previous round of review and you feel that this manuscript is now acceptable for publication, you may indicate that here to bypass the “Comments to the Author” section, enter your conflict of interest statement in the “Confidential to Editor” section, and submit your "Accept" recommendation.

Reviewer #3: All comments have been addressed

2. Is the manuscript technically sound, and do the data support the conclusions?

Reviewer #3: Yes

3. Has the statistical analysis been performed appropriately and rigorously? 

Reviewer #3: Yes

4. Have the authors made all data underlying the findings in their manuscript fully available?

Reviewer #3: Yes

5. Is the manuscript presented in an intelligible fashion and written in standard English?

Reviewer #3: Yes

6. Review Comments to the Author

Reviewer #3: 1. There is a mismatch between Fig 1(B) legend and the text in the Fig 1(B). Please correct the word ‘αKG and NADPH’ to ‘ICT and NADP+’ in the figure legend in order to match the labels.

2. There are some typographical errors in the manuscripts that should be corrected.

Fig 2(A) legend: Lys212' as the bse initiator > Lys212' as the base initiator

Page 3, line 54: anddressed > addressed

Page 5, line 152: pre > per

Page 6, line 189: X-structure > X-ray structure

Page 7, line 204: the the > the

7. PLOS authors have the option to publish the peer review history of their article (what does this mean?). If published, this will include your full peer review and any attached files.

Reviewer #3: No

---

## [Editor Report · Acceptance letter]

PONE-D-24-47203R1

PLOS ONE

Dear Dr. Carloni,

I'm pleased to inform you that your manuscript has been deemed suitable for publication in PLOS ONE. Congratulations! Your manuscript is now being handed over to our production team.

Kind regards,

on behalf of

Dr. Michael Klymkowsky

Academic Editor

PLOS ONE